# An Approximate, Efficient Solver for LP Rounding

**Srikrishna Sridhar**[1], **Victor Bittorf**[1], **Ji Liu**[1], **Ce Zhang**[1]
**Christopher Ré**[2], **Stephen J. Wright**[1]
[1]Computer Sciences Department, University of Wisconsin-Madison, Madison, WI 53706
[2]Computer Science Department, Stanford University, Stanford, CA 94305
{srikris,vbittorf,ji-liu,czhang,swright}@cs.wisc.edu
chrismre@cs.stanford.edu

## Abstract

Many problems in machine learning can be solved by rounding the solution of an appropriate linear program (LP). This paper shows that we can recover solutions of comparable quality by rounding an approximate LP solution instead of the exact one. These approximate LP solutions can be computed efficiently by applying a parallel stochastic-coordinate-descent method to a quadratic-penalty formulation of the LP. We derive worst-case runtime and solution quality guarantees of this scheme using novel perturbation and convergence analysis. Our experiments demonstrate that on such combinatorial problems as vertex cover, independent set and multiway-cut, our approximate rounding scheme is up to an order of magnitude faster than Cplex (a commercial LP solver) while producing solutions of similar quality.

## 1 Introduction

A host of machine-learning problems can be solved effectively as approximations of such NP-hard combinatorial problems as set cover, set packing, and multiway-cuts [8, 11, 16, 22]. A popular scheme for solving such problems is called LP rounding [22, chs. 12-26], which consists of the following three-step process: (1) construct an integer (binary) linear program (IP) formulation of a given problem; (2) relax the IP to an LP by replacing the constraints $x \in \{0, 1\}$ by $x \in [0, 1]$; and (3) round an optimal solution of the LP to create a feasible solution for the original IP problem. LP rounding is known to work well on a range of hard problems, and comes with theoretical guarantees for runtime and solution quality.

The Achilles' heel of LP-rounding is that it requires solutions of LPs of possibly extreme scale. Despite decades of work on LP solvers, including impressive advances during the 1990s, commercial codes such as Cplex or Gurobi may not be capable of handling problems of the required scale. In this work, we propose an approximate LP solver suitable for use in the LP-rounding approach, for very large problems. Our intuition is that in LP rounding, since we ultimately round the LP to obtain an approximate solution of the combinatorial problem, a crude solution of the LP may suffice. Hence, an approach that can find approximate solutions of large LPs quickly may be suitable, even if it is inefficient for obtaining highly accurate solutions.

This paper focuses on the theoretical and algorithmic aspects of finding approximate solutions to an LP, for use in LP-rounding schemes. Our three main technical contributions are as follows: First, we show that one can approximately solve large LPs by forming convex quadratic programming (QP) approximations, then applying stochastic coordinate descent to these approximations. Second, we derive a novel convergence analysis of our method, based on Renegar's perturbation theory for linear programming [17]. Finally, we derive bounds on runtime as well as worst-case approximation ratio of our rounding schemes. Our experiments demonstrate that our approach, called Thetis, produces solutions of comparable quality to state-of-the-art approaches on such tasks as noun-phrase chunking and entity resolution. We also demonstrate, on three different classes of combinatorial problems, that Thetis can outperform Cplex (a state-of-the-art commercial LP and IP solver) by up to an order of magnitude in runtime, while achieving comparable solution quality.

**Related Work.** Recently, there has been some focus on the connection between LP relaxations and maximum a posteriori (MAP) estimation problems [16, 19]. Ravikumar et. al [16] proposed rounding schemes for iterative LP solvers to facilitate MAP inference in graphical models. In contrast, we propose to use stochastic descent methods to solve a QP relaxation; this allows us to take advantage of recent results on asynchronous parallel methods of this type [12, 14]. Recently, Makari et. al [13] propose an intriguing parallel scheme for packing and covering problems. In contrast, our results apply to more general LP relaxations, including set-partitioning problems like multiway-cut. Additionally, the runtime of our algorithm is less sensitive to approximation error. For an error $\varepsilon$, the bound on runtime of the algorithm in [13] grows as $\varepsilon^{-5}$, while the bound on our algorithm's runtime grows as $\varepsilon^{-2}$.

## 2 Background: Approximating NP-hard problems with LP Rounding

In this section, we review the theory of LP-rounding based approximation schemes for NP-hard combinatorial problems. We use the vertex cover problem as an example, as it is the simplest nontrivial setting that exposes the main ideas of this approach.

**Preliminaries.** For a minimization problem $\Phi$, an algorithm ALG is an $\alpha$-*factor approximation for* $\Phi$, for some $\alpha > 1$, if any solution produced by ALG has an objective value at most $\alpha$ times the value of an optimal (lowest cost) solution. For some problems, such as vertex cover, there is a constant-factor approximation scheme ($\alpha = 2$). For others, such as set cover, the value of $\alpha$ can be as large as $O(\log N)$, where $N$ is the number of sets.

An LP-rounding based approximation scheme for the problem $\Phi$ first *constructs* an IP formulation of $\Phi$ which we denote as "$P$". This step is typically easy to perform, but the IP formulation $P$ is, in theory, as hard to solve as the original problem $\Phi$. In this work, we consider applications in which the only integer variables in the IP formulation are binary variables $x \in \{0, 1\}$. The second step in LP rounding is a *relax / solve* step: We relax the constraints in $P$ to obtain a linear program $LP(P)$, replacing the binary variables with continuous variables in $[0, 1]$, then solve $LP(P)$. The third step is to *round* the solution of $LP(P)$ to an integer solution which is feasible for $P$, thus yielding a candidate solution to the original problem $\Phi$. The focus of this paper is on the relax / solve step, which is usually the computational bottleneck in an LP-rounding based approximation scheme.

**Example: An Oblivious-Rounding Scheme For Vertex Cover.** Let $G(V, E)$ denote a graph with vertex set $V$ and undirected edges $E \subseteq (V \times V)$. Let $c_v$ denote a nonnegative cost associated with each vertex $v \in V$. A vertex cover of a graph is a subset of $V$ such that each edge $e \in E$ is incident to at least one vertex in this set. The *minimum-cost* vertex cover is the one that minimizes the sum of terms $c_v$, summed over the vertices $v$ belonging to the cover. Let us review the "construct," "relax / solve," and "round" phases of an LP-rounding based approximation scheme applied to vertex cover.

In the "construct" phase, we introduce binary variables $x_v \in \{0, 1\}$, $\forall v \in V$, where $x_v$ is set to 1 if the vertex $v \in V$ is selected in the vertex cover and 0 otherwise. The IP formulation is as follows:

$$\min_x \sum_{v \in V} c_v x_v \text{ s.t. } x_u + x_v \geq 1 \text{ for } (u, v) \in E \text{ and } x_v \in \{0, 1\} \text{ for } v \in V. \tag{1}$$

Relaxation yields the following LP

$$\min_x \sum_{v \in V} c_v x_v \text{ s.t. } x_u + x_v \geq 1 \text{ for } (u, v) \in E \text{ and } x_v \in [0, 1] \text{ for } v \in V. \tag{2}$$

A feasible solution of the LP relaxation (2) is called a "fractional solution" of the original problem. In the "round" phase, we generate a valid vertex cover by simply choosing the vertices $v \in V$ whose fractional solution $x_v \geq \frac{1}{2}$. It is easy to see that the vertex cover generated by such a rounding scheme costs no more than twice the cost of the fractional solution. If the fractional solution chosen for rounding is an optimal solution of (2), then we arrive at a 2-factor approximation scheme for vertex cover. We note here an important property: The rounding algorithm can generate feasible integral solutions while being *oblivious* of whether the fractional solution is an optimal solution of (2). We formally define the notion of an oblivious rounding scheme as follows.

**Definition 1.** *For a minimization problem $\Phi$ with an IP formulation $P$ whose LP relaxation is denoted by $LP(P)$, a $\gamma$-factor 'oblivious' rounding scheme converts any feasible point $x_f \in LP(P)$ to an integral solution $x_I \in P$ with cost at most $\gamma$ times the cost of $LP(P)$ at $x_f$.*

| Problem Family | Approximation Factor | Machine Learning Applications |
|---|---|---|
| Set Covering | $\log(N)$ [20] | Classification [3], Multi-object tracking [24]. |
| Set Packing | $es + o(s)$ [1] | MAP-inference [19], Natural language [9]. |
| Multiway-cut | $3/2 - 1/k$ [5] | Computer vision [4], Entity resolution [10]. |
| Graphical Models | Heuristic | Semantic role labeling [18], Clustering [21]. |

Figure 1: LP-rounding schemes considered in this paper. The parameter $N$ refers to the number of sets; $s$ refers to $s$-column sparse matrices; and $k$ refers to the number of terminals. $e$ is the Euler's constant.

Given a $\gamma$-factor *oblivious algorithm* ALG to the problem $\Phi$, one can construct a $\gamma$-factor approximation algorithm for $\Phi$ by using ALG to round an *optimal* fractional solution of LP($P$). When we have an approximate solution for LP($P$) that is feasible for this problem, rounding can produce an $\alpha$-factor approximation algorithm for $\Phi$ for a factor $\alpha$ slightly larger than $\gamma$, where the difference between $\alpha$ and $\gamma$ takes account of the inexactness in the approximate solution of LP($P$). Many LP-rounding schemes (including the scheme for vertex cover discussed in Section 2) are oblivious. We implemented the oblivious LP-rounding algorithms in Figure 1 and report experimental results in Section 4.

## 3 Main results

In this section, we describe how we can solve LP relaxations approximately, in less time than traditional LP solvers, while still preserving the formal guarantees of rounding schemes. We first define a notion of approximate LP solution and discuss its consequences for oblivious rounding schemes. We show that one can use a regularized quadratic penalty formulation to compute these approximate LP solutions. We then describe a stochastic-coordinate-descent (SCD) algorithm for obtaining approximate solutions of this QP, and mention enhancements of this approach, specifically, asynchronous parallel implementation and the use of an augmented Lagrangian framework. Our analysis yields a worst-case complexity bound for solution quality and runtime of the entire LP-rounding scheme.

### 3.1 Approximating LP Solutions

Consider the LP in the following standard form

$$\min c^T x \text{ s.t. } Ax = b, \quad x \geq 0, \tag{3}$$

where $c \in \mathbb{R}^n$, $b \in \mathbb{R}^m$, and $A \in \mathbb{R}^{m \times n}$ and its corresponding dual

$$\max b^T u \text{ s.t. } c - A^T u \geq 0. \tag{4}$$

Let $x^*$ denote an optimal primal solution of (3). An approximate LP solution $\hat{x}$ that we use for LP-rounding may be infeasible and have objective value different from the optimum $c^T x^*$. We quantify the inexactness in an approximate LP solution as follows.

**Definition 2.** *A point $\hat{x}$ is an $(\epsilon, \delta)$-approximate solution of the LP (3) if $\hat{x} \geq 0$ and there exists constants $\epsilon > 0$ and $\delta > 0$ such that*

$$\|A\hat{x} - b\|_\infty \leq \epsilon \quad \text{and} \quad |c^T \hat{x} - c^T x^*| \leq \delta |c^T x^*|.$$

Using Definitions 1 and 2, it is easy to see that a $\gamma$-factor oblivious rounding scheme can round a $(0, \delta)$ approximate solution to produce a feasible integral solution whose cost is no more than $\gamma(1 + \delta)$ times the optimal solution of the $P$. The factor $(1 + \delta)$ arises because the rounding algorithm does not have access to an optimal fractional solution. To cope with the infeasibility, we convert an $(\epsilon, \delta)$-approximate solution to a $(0, \hat{\delta})$ approximate solution where $\hat{\delta}$ is not too large. For vertex cover (2), we prove the following result in Appendix C. (Here, $\Pi_{[0,1]^n}(\cdot)$ denotes projection onto the unit hypercube in $\mathbb{R}^n$.)

**Lemma 3.** *Let $\hat{x}$ be an $(\varepsilon, \delta)$ approximate solution to the linear program (2) with $\varepsilon \in [0, 1)$. Then, $\tilde{x} = \Pi_{[0,1]^n}((1 - \varepsilon)^{-1}\hat{x})$ is a $(0, \delta(1 - \varepsilon)^{-1})$-approximate solution.*

Since $\tilde{x}$ is a feasible solution for (2), the oblivious rounding scheme in Section 2 results in an $2(1 + \delta(1 - \varepsilon)^{-1})$ factor approximation algorithm. In general, constructing $(0, \hat{\delta})$ from $(\epsilon, \delta)$ approximate solutions requires reasoning about the structure of a particular LP. In Appendix C, we establish statements analogous to Lemma 3 for packing, covering and multiway-cut problems.

## 3.2 Quadratic Programming Approximation to the LP

We consider the following regularized quadratic penalty approximation to the LP (3), parameterized by a positive constant $\beta$, whose solution is denoted by $x(\beta)$:

$$x(\beta) := \arg\min_{x \geq 0} f_\beta(x) := c^T x - \bar{u}^T(Ax - b) + \frac{\beta}{2}\|Ax - b\|^2 + \frac{1}{2\beta}\|x - \bar{x}\|^2, \qquad (5)$$

where $\bar{u} \in \mathbb{R}^m$ and $\bar{x} \in \mathbb{R}^n$ are arbitrary vectors. (In practice, $\bar{u}$ and $\bar{x}$ may be chosen as approximations to the dual and primal solutions of (3), or simply set to zero.) The quality of the approximation (5) depends on the *conditioning* of underlying linear program (3), a concept that was studied by Renegar [17]. Denoting the data for problem (3) by $d := (A, b, c)$, we consider perturbations $\Delta d := (\Delta A, \Delta b, \Delta c)$ such that the linear program defined by $d + \Delta d$ is primal infeasible. The primal condition number $\delta_P$ is the infimum of the ratios $\|\Delta d\|/\|d\|$ over all such vectors $\Delta d$. The dual condition number $\delta_D$ is defined analogously. (Clearly both $\delta_P$ and $\delta_D$ are in the range $[0, 1]$; smaller values indicate poorer conditioning.) We have the following result, which is proven in the supplementary material.

**Theorem 4.** *Suppose that $\delta_P$ and $\delta_D$ are both positive, and let $(x^*, u^*)$ be any primal-dual solution pair for (3), (4). If we define $C_* := \max(\|x^* - \bar{x}\|, \|u^* - \bar{u}\|)$, then the unique solution $x(\beta)$ of (5) satisfies*

$$\|Ax(\beta) - b\| \leq (1/\beta)(1 + \sqrt{2})C_*, \quad \|x(\beta) - x^*\| \leq \sqrt{6}C_*.$$

*If in addition the parameter $\beta \geq \frac{10C_*}{\|d\|\min(\delta_P, \delta_D)}$, then we have*

$$|c^T x^* - c^T x(\beta)| \leq \frac{1}{\beta}\left[\frac{25C_*}{2\delta_P\delta_D} + 6C_*^2 + \sqrt{6}\|\bar{x}\|C_*\right].$$

In practice, we solve (5) approximately, using an algorithm whose complexity depends on the threshold $\bar{\epsilon}$ for which the objective is accurate to within $\bar{\epsilon}$. That is, we seek $\hat{x}$ such that

$$\beta^{-1}\|\hat{x} - x(\beta)\|^2 \leq f_\beta(\hat{x}) - f_\beta(x(\beta)) \leq \bar{\epsilon},$$

where the left-hand inequality follows from the fact that $f_\beta$ is strongly convex with modulus $\beta^{-1}$. If we define

$$\bar{\epsilon} := \frac{C_{20}^2}{\beta^3}, \quad C_{20} := \frac{25C_*}{2\|d\|\delta_P\delta_D}, \qquad (6)$$

then by combining some elementary inequalities with the results of Theorem 4, we obtain the bounds

$$|c^T\hat{x} - c^T x^*| \leq \frac{1}{\beta}\left[\frac{25C_*}{\delta_P\delta_D} + 6C_*^2 + \sqrt{6}\|\bar{x}\|C_*\right], \quad \|A\hat{x} - b\| \leq \frac{1}{\beta}\left[(1 + \sqrt{2})C_* + \frac{25C_*}{2\delta_P\delta_D}\right].$$

The following result is almost an immediate consequence.

**Theorem 5.** *Suppose that $\delta_P$ and $\delta_D$ are both positive and let $(x^*, u^*)$ be any primal-dual optimal pair. Suppose that $C_*$ is defined as in Theorem 4. Then for any given positive pair $(\epsilon, \delta)$, we have that $\hat{x}$ satisfies the inequalities in Definition 2 provided that $\beta$ satisfies the following three lower bounds:*

$$\beta \geq \frac{10C_*}{\|d\|\min(\delta_P, \delta_D)},$$

$$\beta \geq \frac{1}{\delta|c^T x^*|}\left[\frac{25C_*}{\delta_P\delta_D} + 6C_*^2 + \sqrt{6}\|\bar{x}\|C_*\right],$$

$$\beta \geq \frac{1}{\epsilon}\left[(1 + \sqrt{2})C_* + \frac{25C_*}{2\delta_P\delta_D}\right].$$

For an instance of vertex cover with $n$ nodes and $m$ edges, we can show that $\delta_P^{-1} = O(n^{1/2}(m + n)^{1/2})$ and $\delta_D^{-1} = O((m + n)^{1/2})$ (see Appendix D). The values $\bar{x} = \mathbf{1}$ and $\bar{u} = \vec{0}$ yield $C_* \leq \sqrt{m}$. We therefore obtain $\beta = O(m^{1/2}n^{1/2}(m + n)(\min\{\epsilon, \delta|c^T x^*|\})^{-1})$.

---
**Algorithm 1** SCD method for (5)
---
1: Choose $x_0 \in \mathbb{R}^n$; $j \leftarrow 0$
2: **loop**
3:     Choose $i(j) \in \{1, 2, \ldots, n\}$ randomly with equal probability;
4:     Define $x_{j+1}$ from $x_j$ by setting $[x_{j+1}]_{i(j)} \leftarrow \max(0, [x_j]_{i(j)} - (1/L_{\max})[\nabla f_\beta(x_j)]_{i(j)})$, leaving other components unchanged;
5:     $j \leftarrow j + 1$;
6: **end loop**
---

### 3.3 Solving the QP Approximation: Coordinate Descent

We propose the use of a stochastic coordinate descent (SCD) algorithm [12] to solve (5). Each step of SCD chooses a component $i \in \{1, 2, \ldots, n\}$ and takes a step in the $i$th component of $x$ along the partial gradient of (5) with respect to this component, projecting if necessary to retain nonnegativity. This simple procedure depends on the following constant $L_{\max}$, which bounds the diagonals of the Hessian in the objective of (5):

$$L_{\max} = \beta(\max_{i=1,2,\ldots,n} A_{:i}^T A_{:i}) + \beta^{-1}, \tag{7}$$

where $A_{:i}$ denotes the $i$th column of $A$. Algorithm 1 describes the SCD method. Convergence results for Algorithm 1 can be obtained from [12]. In this result, $\mathbb{E}(\cdot)$ denotes expectation over all the random variables $i(j)$ indicating the update indices chosen at each iteration. We need the following quantities:

$$l := \frac{1}{\beta}, \quad R := \sup_{j=1,2,\ldots n} \|x_j - x(\beta)\|_2, \tag{8}$$

where $x_j$ denotes the $j$th iterate of the SCD algorithm. (Note that $R$ bounds the maximum distance that the iterates travel from the solution $x(\beta)$ of (5).)

**Theorem 6.** *For Algorithm 1 we have*

$$\mathbb{E}\|x_j - x(\beta)\|^2 + \frac{2}{L_{\max}}\mathbb{E}(f_\beta(x_j) - f_\beta^*) \leq \left(1 - \frac{l}{n(l + L_{\max})}\right)^j \left(R^2 + \frac{2}{L_{\max}}(f_\beta(x_0) - f_\beta^*)\right),$$

*where $f_\beta^* := f_\beta(x(\beta))$. We obtain high-probability convergence of $f_\beta(x_j)$ to $f_\beta^*$ in the following sense: For any $\eta \in (0,1)$ and any small $\bar{\epsilon}$, we have $P(f_\beta(x_j) - f_\beta^* < \bar{\epsilon}) \geq 1 - \eta$, provided that*

$$j \geq \frac{n(l + L_{\max})}{l}\left|\log \frac{L_{\max}}{2\eta\bar{\epsilon}}\left(R^2 + \frac{2}{L_{\max}}(f_\beta(x_0) - f_\beta^*)\right)\right|.$$

**Worst-Case Complexity Bounds.**   We now combine the analysis in Sections 3.2 and 3.3 to derive a worst-case complexity bound for our approximate LP solver. Supposing that the columns of $A$ have norm $O(1)$, we have from (7) and (8) that $l = \beta^{-1}$ and $L_{\max} = O(\beta)$. Theorem 6 indicates that we require $O(n\beta^2)$ iterations to solve (5) (modulo a log term). For the values of $\beta$ described in Section 3.2, this translates to a complexity estimate of $O(m^3 n^2/\epsilon^2)$.

In order to obtain the desired accuracy in terms of feasibility and function value of the LP (captured by $\epsilon$) we need to solve the QP to within the different, tighter tolerance $\bar{\epsilon}$ introduced in (6). Both tolerances are related to the choice of penalty parameter $\beta$ in the QP. Ignoring here the dependence on dimensions $m$ and $n$, we note the relationships $\beta \sim \epsilon^{-1}$ (from Theorem 5) and $\bar{\epsilon} \sim \beta^{-3} \sim \epsilon^3$ (from (6)). Expressing all quantities in terms of $\epsilon$, and using Theorem 6, we see an iteration complexity of $\epsilon^{-2}$ for SCD (ignoring log terms). The linear convergence rate of SCD is instrumental to this favorable value. By contrast, standard variants of stochastic-gradient descent (SGD) applied to the QP yield poorer complexity. For diminishing-step or constant-step variants of SGD, we see complexity of $\epsilon^{-7}$, while for robust SGD, we see $\epsilon^{-10}$. (Besides the inverse dependence on $\bar{\epsilon}$ or its square in the analysis of these methods, there is a contribution of order $\epsilon^{-2}$ from the conditioning of the QP.)

### 3.4 Enhancements

We mention two important enhancements that improve the efficiency of the approach outlined above. The first is an asynchronous parallel implementation of Algorithm 1 and the second is the use of an augmented Lagrangian framework rather than "one-shot" approximation by the QP in (5).

| | | | | Thetis | | | | Gibbs Sampling | | | |
|---|---|---|---|---|---|---|---|---|---|---|---|
| Task | Formulation | PV | NNZ | P | R | F1 | Rank | P | R | F1 | Rank |
| CoNLL | Skip-chain CRF | 25M | 51M | .87 | .90 | .89 | 10/13 | .86 | .90 | .88 | 10/13 |
| TAC-KBP | Factor graph | 62K | 115K | .79 | .79 | .79 | 6/17 | .80 | .80 | .80 | 6/17 |

Figure 2: Solution quality of our LP-rounding approach on two tasks. PV is the number of primal variables and NNZ is the number of non-zeros in the constraint matrix of the LP in standard form. The *rank* indicates where we would been have placed, had we participated in the competition.

**Asynchronous Parallel SCD.** An asynchronous parallel version of Algorithm 1, described in [12], is suitable for execution on multicore, shared-memory architectures. Each core, executing a single thread, has access to the complete vector $x$. Each thread essentially runs its own version of Algorithm 1 independently of the others, choosing and updating one component $i(j)$ of $x$ on each iteration. Between the time a thread reads $x$ and performs its update, $x$ usually will have been updated by several other threads. Provided that the number of threads is not too large (according to criteria that depends on $n$ and on the diagonal dominance properties of the Hessian matrix), and the step size is chosen appropriately, the convergence rate is similar to the serial case, and near-linear speedup is observed.

**Augmented Lagrangian Framework.** It is well known (see for example [2, 15]) that the quadratic-penalty approach can be extended to an augmented Lagrangian framework, in which a sequence of problems of the form (5) are solved, with the primal and dual solution estimates $\bar{x}$ and $\bar{u}$ (and possibly the penalty parameter $\beta$) updated between iterations. Such a "proximal method of multipliers" for LP was described in [23]. We omit a discussion of the convergence properties of the algorithm here, but note that the quality of solution depends on the values of $\bar{x}$, $\bar{u}$ and $\beta$ at the last iteration before convergence is declared. By applying Theorem 5, we note that the constant $C_*$ is smaller when $\bar{x}$ and $\bar{u}$ are close to the primal and dual solution sets, thus improving the approximation and reducing the need to increase $\beta$ to a larger value to obtain an approximate solution of acceptable accuracy.

# 4 Experiments

Our experiments address two main questions: (1) Is our approximate LP-rounding scheme useful in graph analysis tasks that arise in machine learning? and (2) How does our approach compare to a state-of-the-art commercial solver? We give favorable answers to both questions.

## 4.1 Is Our Approximate LP-Rounding Scheme Useful in Graph Analysis Tasks?

LP formulations have been used to solve MAP inference problems on graphical models [16], but general-purpose LP solvers have rarely been used, for reasons of scalability. We demonstrate that the rounded solutions obtained using Thetis are of comparable quality to those obtained with state-of-the-art systems. We perform experiments on two different tasks: entity linking and text chunking. For each task, we produce a factor graph [9], which consists of a set of random variables and a set of factors to describe the correlation between random variables. We then run MAP inference on the factor graph using the LP formulation in [9] and compare the quality of the solutions obtained by Thetis with a Gibbs sampling-based approach [26]. We follow the LP-rounding algorithm in [16] to solve the MAP estimation problem. For entity linking, we use the TAC-KBP 2010 benchmark[1]. The input graphical model has 12K boolean random variables and 17K factors. For text chunking, we use the CoNLL 2000 shared task[2]. The factor graph contained 47K categorical random variables (with domain size 23) and 100K factors. We use the training sets provided by TAC-KBP 2010 and CoNLL 2000 respectively. We evaluate the quality of both approaches using the official evaluation scripts and evaluation data sets provided by each challenge. Figure 2 contains a description of the three relevant quality metrics, precision (P), recall (R) and F1-scores. Figure 2 demonstrates that our algorithm produces solutions of quality comparable with state-of-the-art approaches for these graph analysis tasks.

## 4.2 How does our proposed approach compare to a state-of-the-art commercial solver?

We conducted numerical experiments on three different combinatorial problems that commonly arise in graph analysis tasks in machine learning: vertex cover, independent set, and multiway cuts. For

each problem, we compared the performance of our LP solver against the LP and IP solvers of Cplex (v12.5) (denoted as Cplex-LP and Cplex-IP respectively). The two main goals of this experiment are to: (1) compare the quality of the integral solutions obtained using LP-rounding with the integral solutions from Cplex-IP and (2) compare wall-clock times required by Thetis and Cplex-LP to solve the LPs for the purpose of LP-rounding.

**Datasets.** Our tasks are based on two families of graphs. The first family of instances (*frb59-26-1* to *frb59-26-5*) was obtained from Bhoslib[3] (Benchmark with Hidden Optimum Solutions); they are considered difficult problems [25]. The instances in this family are similar; the first is reported in the figures of this section, while the remainder appear in Appendix E. The second family of instances are social networking graphs obtained from the Stanford Network Analysis Platform (SNAP)[4].

**System Setup.** Thetis was implemented using a combination of C++ (for Algorithm 1) and Matlab (for the augmented Lagrangian framework). Our implementation of the augmented Lagrangian framework was based on [6]. All experiments were run on a 4 Intel Xeon E7-4450 (40 cores @ 2Ghz) with 256GB of RAM running Linux 3.8.4 with a 15-disk RAID0. Cplex used 32 (of the 40) cores available in the machine, and for consistency, our implementation was also restricted to 32 cores. Cplex implements presolve procedures that detect redundancy, and substitute and eliminate variables to obtain equivalent, smaller LPs. Since the aim of this experiment is compare the algorithms used to solve LPs, we ran both Cplex-LP and Thetis on the reduced LPs generated by the presolve procedure of Cplex-LP. Both Cplex-LP and Thetis were run to a tolerance of $\epsilon = 0.1$. Additional experiments with Cplex-LP run using its default tolerance options are reported in Appendix E. We used the barrier optimizer while running Cplex-LP. All codes were provided with a time limit of 3600 seconds excluding the time taken for preprocessing as well as the runtime of the rounding algorithms that generate integral solutions from fractional solutions.

**Tasks.** We solved the vertex cover problem using the approximation algorithm described in Section 2. We solved the maximum independent set problem using a variant of the $es + o(s)$-factor approximation in [1] where $s$ is the maximum degree of a node in the graph (see Appendix C for details). For the multiway-cut problem (with $k = 3$) we used the $3/2 - 1/k$-factor approximation algorithm described in [22]. The details of the transformation from approximate infeasible solutions to feasible solutions are provided in Appendix C. Since the rounding schemes for maximum-independent set and multiway-cut are randomized, we chose the best feasible integral solution from 10 repetitions.

| Instance | Minimization problems | | | | | | | | Maximization problems | | | |
|---|---|---|---|---|---|---|---|---|---|---|---|---|
| | VC | | | | MC | | | | MIS | | | |
| | PV | NNZ | S | Q | PV | NNZ | S | Q | PV | NNZ | S | Q |
| frb59-26-1 | 0.12 | 0.37 | 2.8 | 1.04 | 0.75 | 3.02 | 53.3 | 1.01 | 0.12 | 0.38 | 5.3 | 0.36 |
| Amazon | 0.39 | 1.17 | 8.4 | 1.23 | 5.89 | 23.2 | - | 0.42 | 0.39 | 1.17 | 7.4 | 0.82 |
| DBLP | 0.37 | 1.13 | 8.3 | 1.25 | 6.61 | 26.1 | - | 0.33 | 0.37 | 1.13 | 8.5 | 0.88 |
| Google+ | 0.71 | 2.14 | 9.0 | 1.21 | 9.24 | 36.8 | - | 0.83 | 0.71 | 2.14 | 10.2 | 0.82 |

Figure 3: Summary of wall-clock speedup (in comparison with Cplex-LP) and solution quality (in comparison with Cplex-IP) of Thetis on three graph analysis problems. Each code is run with a time limit of one hour and parallelized over 32 cores, with '-' indicating that the code reached the time limit. PV is the number of primal variables while NNZ is the number of nonzeros in the constraint matrix of the LP in standard form (both in millions). S is the speedup, defined as the time taken by Cplex-LP divided by the time taken by Thetis. Q is the ratio of the solution objective obtained by Thetis to that reported by Cplex-IP. For minimization problems (**VC** and **MC**) lower Q is better; for maximization problems (**MIS**) higher Q is better. For MC, a value of $Q < 1$ indicates that Thetis found a better solution than Cplex-IP found within the time limit.

**Results.** The results are summarized in Figure 3, with additional details in Figure 4. We discuss the results for the vertex cover problem. On the Bhoslib instances, the integral solutions from Thetis were within 4% of the documented optimal solutions. In comparison, Cplex-IP produced

| VC | Cplex IP | | | Cplex LP | | | Thetis | | |
|---|---|---|---|---|---|---|---|---|---|
| (min) | t (secs) | BFS | Gap (%) | t (secs) | LP | RSol | t (secs) | LP | RSol |
| frb59-26-1 | - | 1475 | 0.67 | 2.48 | 767 | 1534 | 0.88 | 959.7 | 1532 |
| Amazon | 85.5 | $1.60\times10^5$ | - | 24.8 | $1.50\times10^5$ | $2.04\times10^5$ | 2.97 | $1.50\times10^5$ | $1.97\times10^5$ |
| DBLP | 22.1 | $1.65\times10^5$ | - | 22.3 | $1.42\times10^5$ | $2.08\times10^5$ | 2.70 | $1.42\times10^5$ | $2.06\times10^5$ |
| Google+ | - | $1.06\times10^5$ | 0.01 | 40.1 | $1.00\times10^5$ | $1.31\times10^5$ | 4.47 | $1.00\times10^5$ | $1.27\times10^5$ |
| **MC** | Cplex IP | | | Cplex LP | | | Thetis | | |
| (min) | t (secs) | BFS | Gap (%) | t (secs) | LP | RSol | t (secs) | LP | RSol |
| frb59-26-1 | 72.3 | 346 | - | 312.2 | 346 | 346 | 5.86 | 352.3 | 349 |
| Amazon | - | 12 | NA | - | - | - | 55.8 | 7.28 | 5 |
| DBLP | - | 15 | NA | - | - | - | 63.8 | 11.7 | 5 |
| Google+ | - | 6 | NA | - | - | - | 109.9 | 5.84 | 5 |
| **MIS** | Cplex IP | | | Cplex LP | | | Thetis | | |
| (max) | t (secs) | BFS | Gap (%) | t (secs) | LP | RSol | t (secs) | LP | RSol |
| frb59-26-1 | - | 50 | 18.0 | 4.65 | 767 | 15 | 0.88 | 447.7 | 18 |
| Amazon | 35.4 | $1.75\times10^5$ | - | 23.0 | $1.85\times10^5$ | $1.56\times10^5$ | 3.09 | $1.73\times10^5$ | $1.43\times10^5$ |
| DBLP | 17.3 | $1.52\times10^5$ | - | 23.2 | $1.75\times10^5$ | $1.41\times10^5$ | 2.72 | $1.66\times10^5$ | $1.34\times10^5$ |
| Google+ | - | $1.06\times10^5$ | - | 44.5 | $1.11\times10^5$ | $9.39\times10^4$ | 4.37 | $1.00\times10^5$ | $8.67\times10^4$ |

Figure 4: Wall-clock time and quality of fractional and integral solutions for three graph analysis problems using Thetis, Cplex-IP and Cplex-LP. Each code was given a time limit of one hour, with '-' indicating a timeout. BFS is the objective value of the best integer feasible solution found by Cplex-IP. The gap is defined as (BFS−BB)/BFS where BB is the best known solution bound found by Cplex-IP within the time limit. A gap of '-' indicates that the problem was solved to within $0.01\%$ accuracy and NA indicates that Cplex-IP was unable to find a valid solution bound. LP is the objective value of the LP solution, and RSol is objective value of the rounded solution.

integral solutions that were within 1% of the documented optimal solutions, but required an hour for each of the instances. Although the LP solutions obtained by Thetis were less accurate than those obtained by Cplex-LP, the rounded solutions from Thetis and Cplex-LP are almost exactly the same. In summary, the LP-rounding approaches using Thetis and Cplex-LP obtain integral solutions of comparable quality with Cplex-IP — but Thetis is about three times faster than Cplex-LP.

We observed a similar trend on the large social networking graphs. We were able to recover integral solutions of comparable quality to Cplex-IP, but seven to eight times faster than using LP-rounding with Cplex-LP. We make two additional observations. The difference between the optimal fractional and integral solutions for these instances is much smaller than frb59-26-1. We recorded unpredictable performance of Cplex-IP on large instances. Notably, Cplex-IP was able to find the optimal solution for the *Amazon* and *DBLP* instances, but timed out on *Google+*, which is of comparable size. On some instances, Cplex-IP outperformed even Cplex-LP in wall clock time, due to specialized presolve strategies.

# 5    Conclusion

We described Thetis, an LP rounding scheme based on an approximate solver for LP relaxations of combinatorial problems. We derived worst-case runtime and solution quality bounds for our scheme, and demonstrated that our approach was faster than an alternative based on a state-of-the-art LP solver, while producing rounded solutions of comparable quality.

# Acknowledgements

SS is generously supported by ONR award N000141310129. JL is generously supported in part by NSF awards DMS-0914524 and DMS-1216318 and ONR award N000141310129. CR's work on this project is generously supported by NSF CAREER award under IIS-1353606, NSF award under CCF-1356918, the ONR under awards N000141210041 and N000141310129, a Sloan Research Fellowship, and gifts from Oracle and Google. SJW is generously supported in part by NSF awards DMS-0914524 and DMS-1216318, ONR award N000141310129, DOE award DE-SC0002283, and Subcontract 3F-30222 from Argonne National Laboratory. Any recommendations, findings or opinions expressed in this work are those of the authors and do not necessarily reflect the views of any of the above sponsors.

## Footnotes

[1] http://nlp.cs.qc.cuny.edu/kbp/2010/

[2] http://www.cnts.ua.ac.be/conll2000/chunking/

[3]http://www.nlsde.buaa.edu.cn/~kexu/benchmarks/graph-benchmarks.htm

[4]http://snap.stanford.edu/

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
