[Supplementary Material]

# Supplementary Material

## A   Perturbation Results

We discuss here the perturbation results for the quadratic approximation (5) to the linear program (3). These results constitute a proof of Theorem 5.

We note for future reference that the optimality (KKT) conditions for the primal-dual pair of LPs (3) and (4) are

$$Ax = b, \quad 0 \le c - A^T u \perp x \ge 0. \tag{9}$$

The QP approximation (5) is equivalent to the following monotone linear complementarity problem (LCP):

$$0 \le x \perp F_\beta(x) := c - A^T \bar{u} + \beta A^T (Ax - b) + \frac{1}{\beta}(x - \bar{x}). \tag{10}$$

Here we rely on Renegar's theory [17] which requires not only that primal and dual are both solvable, but also that they are still solvable after we make arbitrary small perturbations to the data $(A, b, c)$. This includes cases in which the basis has fewer nonzeros than there are equality constraints (a situation known as "primal degeneracy"). We assume throughout that $A$ has full row rank $m$. If $A$ were row rank deficient, then even if the primal-dual pair had a solution, we would be able to find an arbitrarily small perturbation that renders the primal infeasible.

In accordance with Renegar, we use $d := (A, b, c)$ to denote the data for the problems (3) and (4). We denote by Pri$\emptyset$ the set of data $d$ for which the primal (3) is infeasible, and define Dual$\emptyset$ analogously for the dual (4). Renegar uses the "distance to infeasibility" to define a condition number for the primal and dual. Specifically, defining

$$\delta_P := \frac{\text{dist}(d, \text{Pri}\emptyset)}{\|d\|}, \quad \delta_D := \frac{\text{dist}(d, \text{Dual}\emptyset)}{\|d\|}, \tag{11}$$

the quantities $1/\delta_P$ and $1/\delta_D$ capture the sensitivity of the optimal objective value for the problem (3) to perturbations in $b$ and $c$. Note that both $\delta_P$ and $\delta_D$ lie in the interval $[0, 1]$.

We assume $\delta_P > 0$ and $\delta_D > 0$ throughout the analysis below. This implies that the primal and dual are both feasible, hence by strong duality both have solutions $x^*$ and $u^*$ (not necessarily unique).

**Lemma 7.** *Suppose that $\delta_P > 0$ and $\delta_D > 0$, and let $x^*$ be any solution of (3) and $u^*$ be any solution of (4), and define*

$$C_* := \max(\|x^* - \bar{x}\|, \|u^* - \bar{u}\|).$$

*Then the unique solution $x(\beta)$ of (5) satisfies the following inequalities:*

$$\|Ax(\beta) - b\| \le \beta^{-1} \left[ \|u^* - \bar{u}\| + \sqrt{\|u^* - \bar{u}\|^2 + \|x^* - \bar{x}\|^2} \right]$$

$$\le \beta^{-1}(1 + \sqrt{2})C_*,$$

$$\|x(\beta) - \bar{x}\| \le \left[ 2\|u^* - \bar{u}\| \left[ \|u^* - \bar{u}\| + \sqrt{\|u^* - \bar{u}\|^2 + \|x^* - \bar{x}\|^2} \right] + \|x^* - \bar{x}\|^2 \right]^{1/2}$$

$$\le \sqrt{6}C_*.$$

*Proof.* Note that $x^*$ is a feasible point for (5), so we have by optimality of $x(\beta)$ that $f_\beta(x(\beta)) \le f_\beta(x^*)$, that is,

$$c^T x(\beta) - \bar{u}^T (Ax(\beta) - b) + \frac{\beta}{2}\|Ax(\beta) - b\|^2 + \frac{1}{2\beta}\|x(\beta) - \bar{x}\|^2 \le c^T x^* + \frac{1}{2\beta}\|x^* - \bar{x}\|^2,$$

and thus

$$\frac{\beta}{2}\|Ax(\beta) - b\|^2 + \frac{1}{2\beta}\|x(\beta) - \bar{x}\|^2 \le c^T(x^* - x(\beta)) + \bar{u}^T(Ax(\beta) - b) + \frac{1}{2\beta}\|x^* - \bar{x}\|^2.$$

Note from $x(\beta) \ge 0$ and (9) that

$$0 \le x(\beta)^T (c - A^T u^*) \quad \Rightarrow \quad -c^T x(\beta) \le -(u^*)^T Ax(\beta).$$

We also have from (9) that $c^T x^* = (u^*)^T A x^*$. By combining these observations, we obtain

$$\frac{\beta}{2}\|Ax(\beta) - b\|^2 + \frac{1}{2\beta}\|x(\beta) - \bar{x}\|^2 \le (u^* - \bar{u})^T A(x^* - x(\beta)) + \frac{1}{2\beta}\|x^* - \bar{x}\|^2. \quad (12)$$

By dropping the second term on the left-hand side of this expression, multiplying by $\beta$, and using Cauchy-Schwartz and $Ax^* = b$, we obtain

$$\frac{\beta^2}{2}\|Ax(\beta) - b\|^2 \le \|u^* - \bar{u}\|\beta\|Ax(\beta) - b\| + \frac{1}{2}\|x^* - \bar{x}\|^2.$$

Denoting $e_\beta := \beta\|Ax(\beta) - b\|$, this inequality reduces to the condition

$$\frac{1}{2}e_\beta^2 - \|u^* - \bar{u}\|e_\beta - \frac{1}{2}\|x^* - \bar{x}\|^2 \le 0.$$

Solving this quadratic for $e_\beta$, we obtain

$$e_\beta \le \|u^* - \bar{u}\| + \sqrt{\|u^* - \bar{u}\|^2 + \|x^* - \bar{x}\|^2},$$

proving the first claim.

For the second claim, we return to (12), dropping the first term on the left-hand side, to obtain

$$\frac{1}{2\beta}\|x(\beta) - \bar{x}\|^2 \le \|u^* - \bar{u}\|\|Ax(\beta) - b\| + \frac{1}{2\beta}\|x^* - \bar{x}\|^2.$$

By substituting the bound on $\|Ax(\beta) - b\|$ just derived, multiplying by $2\beta$ and taking the square root, we obtain the result. $\qquad\square$

Fixing $\beta$ and $x(\beta)$, we now consider the following perturbed linear program

$$\min c_\beta^T x \quad \text{s.t.} \quad Ax = b_\beta, \quad x \ge 0, \quad (13)$$

and its dual

$$\max b_\beta^T u \quad \text{s.t.} \quad A^T u \le c_\beta, \quad (14)$$

where

$$b_\beta := Ax(\beta), \quad c_\beta := c + \frac{1}{\beta}(x(\beta) - \bar{x}).$$

From Lemma 7, we have

$$\|b - b_\beta\| \le \frac{1}{\beta}(1 + \sqrt{2})C_* \le \frac{2.5C_*}{\beta}, \quad \|c - c_\beta\| \le \frac{1}{\beta}\sqrt{6}C_* \le \frac{2.5C_*}{\beta}. \quad (15)$$

KKT conditions for (13), (14) are

$$0 \le \hat{x} \perp c + \frac{1}{\beta} - A^T \hat{u} \ge 0, \quad A\hat{x} = Ax(\beta).$$

It is easy to check, by comparing with (10), that these conditions are satisfied by

$$\hat{x} = x(\beta), \quad \hat{u} = \bar{u} - \beta A(x(\beta) - x^*).$$

Hence $\hat{x} = x(\beta)$ is a solution of (13). There may be other solutions, but they will have the same objective value, of course.

We now use the following result, which follows immediately from [17, Theorem 1, part (5)].[5]

**Theorem 8.** *Let $d = (A, b, c)$ be the data defining the primal-dual pair (3) and (4), and suppose that $\delta_P$ and $\delta_D$ defined by (11) are both positive. Consider the following perturbation applied to the $b$ and $c$ components: $\Delta d := (0, \Delta b, \Delta c)$, and assume that*

$$\frac{\|\Delta d\|}{\|d\|} \le \delta_P, \quad \frac{\|\Delta d\|}{\|d\|} \le \delta_D.$$

*Then, denoting the solution of (3) by $x^*$ and the solution of the linear program with perturbed data $d + \Delta d$ by $x_\Delta^*$, we have*

$$|c^T x^* - (c + \Delta c)^T x_\Delta^*| \le \frac{\|\Delta b\|}{\delta_D} \frac{\|c\| + \|\Delta c\|}{\text{dist}(d, \text{Pri}\emptyset) - \|\Delta d\|} + \frac{\|\Delta c\|}{\delta_P} \frac{\|b\| + \|\Delta b\|}{\text{dist}(d, \text{Dual}\emptyset) - \|\Delta d\|}.$$

Our main theorem is obtained by applying this result with the perturbations

$$\Delta b := b_\beta - b = Ax(\beta) - b, \quad \Delta c := c_\beta - c = \frac{1}{\beta}(x(\beta) - \bar{x}). \tag{16}$$

We have the following result.

**Theorem 9.** *Suppose that*

$$\beta \geq \bar{\beta} := \frac{10C_*}{\|d\|\min(\delta_P, \delta_D)}.$$

*We then have the following bound on the difference between the optimal values of (3) and (13):*

$$|c^T x^* - c_\beta^T x(\beta)| \leq \frac{1}{\beta}\frac{25C_*}{2\delta_P\delta_D}.$$

*Proof.* Note first that from (15) and

$$\|\Delta d\| \leq \|\Delta b\| + \|\Delta c\| \leq \frac{5C_*}{\beta}.$$

From our assumption on $\beta$, we have

$$\frac{\|\Delta d\|}{\|d\|} \leq \frac{5C_*}{\beta\|d\|} \leq \frac{1}{2}\min(\delta_P, \delta_D),$$

so that the assumptions of Theorem 8 are satisfied. We have moreover from the definitions (11) that

$$\operatorname{dist}(d, \mathrm{Pri}\emptyset) - \|\Delta d\| = \|d\|\left[\delta_P - \frac{\|\Delta d\|}{\|d\|}\right] \geq \frac{1}{2}\|d\|\delta_P,$$

and similarly $\operatorname{dist}(d, \mathrm{Dual}\emptyset) \geq (1/2)\|d\|\delta_D$. By substituting into the inequality of Theorem 8, and using the bounds just derived together with (15), we obtain

$$|c^* x^* - c_\beta^T x(\beta)| \leq \frac{2.5\beta^{-1}C_*}{\delta_D}\frac{(\|c\| + 2.5\beta^{-1}C_*)}{.5\|d\|\delta_P} + \frac{2.5\beta^{-1}C_*}{\delta_P}\frac{(\|b\| + 2.5\beta^{-1}C_*)}{.5\|d\|\delta_D}.$$

Since

$$\|c\| \leq \|d\|, \quad \|b\| \leq \|d\|, \quad \frac{2.5C_*}{\beta} \leq \frac{1}{4}\min(\delta_P, \delta_D)\|d\| \leq \frac{1}{4}\|d\|,$$

we have

$$|c^* x^* - c_\beta^T x(\beta)| \leq \frac{2.5\beta^{-1}C_*(2.5)\|d\|}{(1/2)\|d\|\delta_P\delta_D} = \frac{1}{\beta}\frac{25C_*}{2\delta_P\delta_D},$$

completing the proof. □

The following corollary is almost immediate.

**Corollary 10.** *Suppose the conditions of Theorem 9 are satisfied. Then*

$$|c^T x^* - c^T x(\beta)| \leq \frac{1}{\beta}\left[\frac{25C_*}{2\delta_P\delta_D} + 6C_*^2 + \sqrt{6}\|\bar{x}\|C_*\right].$$

*Proof.* We have from the definition of $c_\beta$ that

$$|c^T x^* - c^T x(\beta)| \leq |c^T x^* - c_\beta^T x(\beta)| + \frac{1}{\beta}x(\beta)^T(x(\beta) - \bar{x})$$

$$= |c^T x^* - c_\beta^T x(\beta)| + \frac{1}{\beta}\|x(\beta) - \bar{x}\|^2 + \frac{1}{\beta}\bar{x}^T(x(\beta) - \bar{x})$$

$$\leq \frac{1}{\beta}\left[\frac{25C_*}{2\delta_P\delta_D} + 6C_*^2 + \sqrt{6}\|\bar{x}\|C_*\right].$$

where the final inequality follow from Lemma 7 and Theorem 9. □

# B    Details of Rounding Schemes

In this section, we provide details of known LP-rounding schemes for covering, packing and multiway-cut problems. (Vazirani [22] provides a comprehensive survey on the theory and algorithms for LP-rounding.). We then discuss how these algorithms can be extended to round $(\epsilon, \delta)$ optimal solutions.

## B.1    Set Cover

Given a universe $U$ with $N$ elements, a collection of subsets $\mathcal{S} = \{S_1, S_2 \ldots S_k\}$ each associated with a positive cost function $c : S \to \mathbb{R}^+$. In the set cover problem, we must identify a minimum cost sub-collection of sets $S$ that covers all elements in $U$. The set cover problem can be formulated as the following IP:

$$\min \sum_{s \in \mathcal{S}} c_s x_s \quad \text{subject to} \quad \sum_{s:a \in s} x_s \geq 1 \ \forall a \in U, \ x_s \in \{0, 1\} \ \forall s \in \mathcal{S}. \tag{17}$$

A simple way to convert a solution $x_s^*$ of the LP relaxation to an integral solution is to pick all sets $x_s$ where $x_s^* > 1/f$, where $f$ is a bound on the maximum number of sets in which a single element is present. Such an algorithm achieves an $f$-factor approximation [7]. An alternative approximation scheme is a randomized scheme due to [20]. In this scheme, we put $s \in \mathcal{S}$ into the set cover with probability equal to the optimal fractional solution $x_s^*$. In expectation, this approximation scheme is a $O(\log N)$-factor approximation, and is a valid set cover with probability $1/2$.

## B.2    Set Packing

Using the same notation for $U$, $N$, $\mathcal{S}$, and $x_s$, $\forall s \in \mathcal{S}$ as above, the set packing problem is to identify the lowest cost collection of mutually disjoint sets. It can be formulated as the following IP:

$$\max \sum_{s \in \mathcal{S}} c_s x_s \quad \text{subject to} \quad \sum_{s:a \in s} w_{a,s} x_s \leq 1 \ \forall a \in U, \ x_s \in \{0, 1\} \ \forall s \in \mathcal{S}, \tag{18}$$

where $w_{a,s}$ is the weight of element $a \in U$ in set $s \in \mathcal{S}$.

Bansal et al. [1] proposed an $ek + o(k)$-factor approximation (see Algorithm B.2) for the special case of $k$-column sparse set packing where the maximum number of sets containing each element is at most $k$. They use the following stronger formulation of the set packing problem:

$$\max \sum_{s \in \mathcal{S}} c_s x_s \tag{19}$$

$$\text{subject to} \quad \sum_{s:a \in s} w_{a,s} x_s \leq 1 \quad \forall a \in U,$$

$$\sum_{a \in B(s)} x_s \leq 1 \quad \forall a \in U,$$

$$x_s \in \{0, 1\} \quad \forall s \in \mathcal{S},$$

where $w_{a,s} = 1$ if the element $a \in U$ is present in set $s \in S$, $c_s$ is the cost of set $s \in S$ and $B(s) := \{a \in U | w_{a,s} > 1/2\}$.

---

**Algorithm 2** A $ek + o(k)$-factor randomized LP-rounding algorithm for set packing

1: Find any feasible solution $\hat{x}$ to the LP relaxation of (19).
2: Choose set $s \in \mathcal{S}$ with probability $\hat{x}_s/(k\theta)$. Let $\mathcal{C} \subseteq \mathcal{S}$ denote the chosen sets.
3: For each set $s \in \mathcal{C}$ and element $a \in U$, let $E_{a,s}$ denote the event that the sets $\{s_2 \in \mathcal{C} : w_{a,s_2} > w_{a,s}\}$ have a total weight (with respect to element $a$) exceeding 1. Mark $s$ for deletion if $E_{a,s}$ occurs for any $a \in s$.
4: Delete all sets from $s \in \mathcal{C}$ that are marked for deletion.

---

## B.3    Multiway-Cuts

Given a graph $G(V, E)$ and a set of terminals $V_1, V_2, \ldots V_k$, a $k$-way cut partitions the set of vertices $V$ into $k$ mutually disjoint sets. The cost of the $k$-way cut is the sum of the costs of all the edges that

run across the partitions. A $k$-way cut of minimum cost is the solution to the following problem:

$$\min \frac{1}{2} \sum_{u,v \in E} c_{u,v} \sum_{i=1}^{k} |x_u^i - x_v^i| \qquad (20)$$

$$\text{subject to} \quad x_v \in \Delta_k \quad \forall v \in V$$

$$x_v \in \{0,1\}^k \quad \forall v \in V,$$

where $\Delta_k := \{x \in \mathbb{R}^k : \sum_{i=1}^{k} x_i = 1, \; x \geq 0\}$ is the set of simplex constraints in $k$ dimensions. Although it might appear that the formulation in (20) is non-linear, one can easily linearize (20) to

$$\min \frac{1}{2} \sum_{u,v \in E} c_{u,v} \sum_{i=1}^{k} x_{uv}^i$$

$$\text{subject to} \quad x_v \in \Delta_k \qquad\qquad \forall v \in V$$

$$x_{uv}^i \geq x_v^i - x_u^i \qquad \forall u, v \in E, i \in \{1 \ldots k\}$$

$$x_{uv}^i \geq x_u^i - x_v^i \qquad \forall u, v \in E, i \in \{1 \ldots k\}$$

$$x_{uv}^i \in [0,1] \qquad\qquad \forall u, v \in E, i \in \{1 \ldots k\}$$

$$x_v^i \in \{0,1\} \qquad\qquad \forall v \in V, i \in \{1 \ldots k\}$$

There is a $3/2 - 1/k$ factor approximation for multiway-cut using the region-growing algorithm due to [5]. The details of the algorithm are laid out in [22, Algorithm 19.4].

## C  Rounding Infeasible Solutions

In this section, we briefly describe how we can extend known LP-rounding algorithms to infeasible $(\epsilon, \delta)$-approximate solutions. We discuss how one can go from an $(\epsilon, \delta)$-approximate solution to a feasible $(0, f(\epsilon, \delta))$-approximate solution, for some positive function $f(\cdot, \cdot)$. The arguments in this section are based on simple ideas of scaling and projection.

As is the case in the main manuscript, we illustrate our approach using vertex cover. Let $\hat{x}$ be an $(\epsilon, \delta)$-approximate solution of the following vertex cover LP:

$$\min_{x \in [0,1]^n} 1^T x \quad \text{subject to} \; x_i + x_j \geq 1 \text{ for } (i,j) \in E,$$

so that in particular, $x_i \in [0,1]$ for all $i$, and $x_i + x_j \geq 1 - \epsilon$ for all $(i,j) \in E$. We claim that the point

$$z := \Pi_{[0,1]^n}(x/(1-\epsilon))$$

is a $(0, \delta/(1-\varepsilon))$-approximate solution. To check feasibility, suppose for contradiction that $z_i + z_j < 1$ for some $(i,j) \in E$. We thus have $z_i < 1$ and $z_j < 1$, so that $z_i = x_i/(1-\epsilon)$ and $z_j = x_j/(1-\epsilon)$. Therefore, $z_i + z_j = (x_i + x_j)/(1-\epsilon) \geq 1$, a contradiction.

### C.1  Rounding for Coverings

We consider a covering program $P = (A, b, c)$ with positive integer data, that is, $(A, b, c) \geq 0$ and $A \in \mathbb{Z}^{m \times n}$, $b \in \mathbb{Z}^m$, and $c \in \mathbb{Z}^n$. Suppose that there are also $[0,1]$ bound constraints on each component of $x$. The problem formulation is as follows:

$$\min_{x \in [0,1]^n} c^T x \quad \text{subject to} \; Ax \geq b. \qquad [P(A,b,c)]$$

To obtain a formulation closer to the standard form (3), we can introduce slack variables and write

$$\min_{x \in [0,1]^n, z \in [0,\infty)^m} c^T x \quad \text{subject to} \; Ax - z = b, \; z \geq 0.$$

We can always set $z = \max\{Ax - b, 0\}$ to translate between feasible solutions of the two programs.

The following quantity $q(P)$ defines a minimum infeasibility measure over all infeasible, integral solutions to $P$:

$$q(P) = \min_{j=1,\ldots,m} \min_{x \in \{0,1\}^n : A_{j\cdot} x < b_j} b_j - A_{j\cdot} x,$$

where $A_{j\cdot}$ denotes the $j$th row of $A$. Notice for $q(P) \geq 1$ for any non-trivial covering program $P$, by integrality alone.

**Lemma 11.** *Let $P$ be a covering program with a nonempty solution set. Let $\hat{x}$ be an $(\epsilon, \delta)$-approximate solution of $P$, and suppose that $\epsilon/q(P) \leq 1$. Then there is a $(0, \delta/(1-\alpha))$-approximate solution $\tilde{x}$ defined as*

$$\tilde{x} = \Pi_{[0,1]^n}((1-\alpha)^{-1}\hat{x}),$$

*where $\alpha \in [\epsilon/q(P), 1)$.*

*Proof.* We first show that $\tilde{x}$ is feasible. Without loss of generality, assume that $z_j = \max(A_{j.}\hat{x} - b_j, 0)$ for $j = 1, \ldots, m$. Since $\hat{x}$ is a $(\epsilon, \delta)$ solution, we have $\|A\hat{x} - z - b\|_\infty \leq \epsilon$. With $z$ defined as in our formula, this bound implies that

$$A\hat{x} = b \geq -\epsilon\mathbf{1}, \tag{21}$$

where $\mathbf{1}$ is the all-ones vector in $\mathbb{R}^n$. After scaling by $\hat{x}$ by $(1-\alpha)^{-1}$, some components may exceed 1. Hence, we partition the indices into two sets $\Omega_1 = \{i \mid \hat{x}_i \geq 1-\alpha\}$ and $\Omega_{<1} = \{1, 2, \ldots, n\}\setminus\Omega_1$. For any $\Omega \subseteq [n]$, we define the following projection operator:

$$\pi_\Omega(x) := \begin{cases} x_i & \text{if } i \in \Omega \\ 0 & \text{otherwise.} \end{cases}$$

We can then write $\tilde{x}$ as follows:

$$\tilde{x} = \pi_{\Omega_1}\mathbf{1} + (1-\alpha)^{-1}\pi_{\Omega_{<1}}\hat{x}.$$

Assume for contradiction that $\tilde{x}$ is infeasible. Then there must be some constraint $j$ for which $A_{j.}\tilde{x} < b_j$. Using the decomposition above and the fact that $\alpha \in (0,1)$, we have

$$A_{j.}\pi_{\Omega_{<1}}\hat{x} < (b_j - A_{j.}\pi_{\Omega_1}\mathbf{1})(1-\alpha). \tag{22}$$

On the other hand, by (21), we have

$$A_{j.}(\pi_{\Omega_1}\mathbf{1} + \pi_{\Omega_{<1}}\hat{x}) \geq A_{j.}(\pi_{\Omega_1}\hat{x} + \pi_{\Omega_{<1}}\hat{x}) \geq b_j - \epsilon.$$

and so

$$A_{j.}\pi_{\Omega_{<1}}\hat{x} \geq b_j - A_{j.}\pi_{\Omega_1}\mathbf{1} - \epsilon \tag{23}$$

By combining (22) and (23), we obtain

$$(b_j - A_{j.}\pi_{\Omega_1}\mathbf{1})(1-\alpha) > (b_j - A_{j.}\pi_{\Omega_1}\mathbf{1}) - \epsilon$$

Since $b_j - A_{j.}\pi_{\Omega_1}\mathbf{1} \geq b_j - A_{j.}\tilde{x} > 0$, we can divide by $b_j - A_{j.}\pi_{\Omega_1}\mathbf{1}$ without changing signs to obtain

$$\frac{\epsilon}{b_j - A_{j.}\pi_{\Omega_1}\mathbf{1}} > \alpha \quad \Rightarrow \quad b_j - A_{j.}\pi_{\Omega_1}\mathbf{1} < \epsilon/\alpha. \tag{24}$$

We have by using the definition of $\alpha$ that $b_j - A_{j.}\pi_{\Omega_1}\mathbf{1} \geq q(P) \geq \epsilon/\alpha$, since $\pi_\Omega\mathbf{1}$ is an integral but infeasible point for (P). This fact contradicts (24), so we have proved feasibility of $\tilde{x}$ for (P).

We now bound the difference between $c^*\tilde{x}$ and $c^*x^*$, where $x^*$ is the optimal solution of (P). Since $\tilde{x}$ is feasible, we have that $c^Tx^* \leq c^T\tilde{x}$. For the upper bound, we have

$$c^T\tilde{x} - c^Tx^* \leq (1-\alpha)^{-1}c^T\hat{x} - c^Tx^* \leq (1-\alpha)^{-1}(c^T\hat{x} - c^Tx^*) \leq \frac{\delta}{1-\alpha}c^Tx^*.$$

The first inequality follows from $c^Tz \geq c^T(\Pi_{[0,1]^n}z)$ since $c \geq 0$; the second inequality is from $\alpha \in (0,1)$; and the third inequality follows from the fact that $\hat{x}$ is a $(\epsilon, \delta)$ approximation. $\qquad \square$

In our experiments, we set $\alpha = \epsilon/q(P)$, which is computed using the approximate $(\epsilon, \delta)$ optimal fractional solution.

## C.2 Rounding for Packing

A packing problem is a maximization linear program $P(A, b, c)$ where $A, b, c \geq 0$ and $A \in \mathbb{Z}^{m \times n}$, $b \in \mathbb{Z}^m$, and $c \in \mathbb{Z}^n$ along with bound constraints $[0, 1]$ on all variables. That is,

$$\max_{u \in [0,1]^m} u^T b \quad \text{subject to} \quad A^T u \leq c. \qquad [P(A, b, c)]$$

In this class of problems, we can assume without loss of generality that $c \geq \mathbf{1}$. The equality constrained formulation of this problem is

$$\max_{u \in [0,1]^m, z \in \mathbb{R}^n} u^T b \quad \text{subject to} \quad A^T u + z = c, \ z \geq 0.$$

(We can set $z = \max(c - A^T u, 0)$ to obtain the equivalence.)

We use $A_{\cdot i}$ to denote the $i$th column of $A$ in the discussion below.

**Lemma 12.** *Let $P$ be a packing program. Let $\hat{u}$ be an $(\epsilon, \delta)$-approximate solution of $P$, then there is a $(0, \frac{\delta + \alpha}{1 + \alpha})$-approximate solution $\tilde{u}$ defined as*

$$\tilde{u} = \hat{u}/(1 + \alpha)$$

*provided that $\hat{u} \in [0, 1]^m$ where $\alpha \geq \epsilon/\left(\min_{i=1,2,\ldots,n} c_i\right)$.*

*Proof.* We observe first that $\tilde{u} \in [0, 1]^m$. To prove that $A^T \tilde{u} \leq c$, note that since $\hat{u}$ is an $(\epsilon, \delta)$-approximate solution, we have

$$A^T \hat{u} \leq c + \epsilon \mathbf{1} \leq c + \alpha \left( \min_{l=1,2,\ldots,n} c_l \right) \mathbf{1} \leq (1 + \alpha)c,$$

proving the claim.

Let $u_*$ be an optimal solution of $P(A, b, c)$. Since $\tilde{u}$ is feasible and this is a maximization problem, we have $u_*^T b \geq \tilde{u}^T b \geq 0$. For the other bound, we have

$$u_*^T b - \tilde{u}^T b = u_*^T b - \frac{1}{1 + \alpha} \hat{u}^T b \leq u_*^T b - \frac{1 - \delta}{1 + \alpha} u_*^T b = \frac{\delta + \alpha}{1 + \alpha} u_*^T b,$$

completing the proof. $\qquad \square$

A quick examination of the proof suggests that we can take $\alpha := \left( \max_{i=1,2,\ldots,n} \frac{A_{\cdot i}^T \hat{u} - c_i}{c_i} \right)_+$, which is never larger than $\alpha$ as defined above. In our experiments, we set $\alpha$ using this tighter bound and $\theta = \frac{1}{k}$ in algorithm B.2. We note that the algorithm is sensitive to the value of $\theta$. Any positive value of $\theta k \geq 1$ will always return a valid independent set. The proofs in [1] require that $\theta$ must be greater or equal to 1, but we found that $\theta = \frac{1}{k}$ works much better in practice.

## C.3 Rounding for Multiway-Cuts

Since we enforce the simplex constraints in the SCD solve, every solution obtained by our quadratic relaxation is automatically feasible for our linear program.

# D Linear Programming Condition Numbers

In this section, we describe estimates of $(\delta_P, \delta_D)$ in detail for vertex cover, and sketch the ideas for estimating these quantities for the other relaxations that we consider in this paper.

## D.1 Vertex Cover: The Bounds in Detail

Consider vertex cover with a graph $G = (V, E)$, where $|V| = n$ and $|E| = m$. The LP relaxation is as follows

$$\min_{x \in \mathbb{R}_+^n} \mathbf{1}^T x \quad \text{subject to} \quad x_v + x_w \geq 1 \text{ for all } (v, w) \in E \text{ and } x_v \leq 1 \text{ for all } v \in V. \qquad (25)$$

The dual of this program is

$$\max_{u \in \mathbb{R}_+^m, z \in \mathbb{R}^+} u^T \mathbf{1} - z^T \mathbf{1} \quad \text{subject to} \quad \sum_{e:e \ni v} u_e - z_v \leq 1 \text{ for each } v \in V.$$

**Computing $\|d\|$.** Define $\|d\| = \max\{\|A\|_F, \|b\|_2, \|c\|_2\}$ for this problem, where $(A, b, c)$ are the data defining (25). We have

$$\|A\|_F = \sqrt{2m + n}, \quad \|b\|_2 = \sqrt{m + n} \quad \|c\| = \sqrt{n}$$

Hence, $\|d\| = \sqrt{2m + n}$.

**Primal Bound.** We define $x = \frac{2}{3}\mathbf{1}$, and figure how large a perturbation $(\Delta A, \Delta b, \Delta c)$ is needed to problem data $(A, b, c)$ to make this particular point infeasible. The norm of this quantity will give a lower bound on the distance to infeasibility.

By construction of $x$, we have that $Ax - b = \frac{1}{3}\mathbf{1}$. For infeasibility with respect to one of the cover constraints, we would need for some $i$ that

$$|(\Delta A)_{i.}x - \Delta b_i| \geq \frac{1}{3},$$

which, given our definition of $x$, would require

$$\frac{2}{3}\sum_{j=1}^{n}|\Delta A_{ij}| + |\Delta b_i| > \frac{1}{3}. \tag{26}$$

We must therefore have that

$$\sum_{j=1}^{n}|\Delta A_{ij}| \geq \frac{1}{4} \quad \text{and/or} \quad |\Delta b_i| > \frac{1}{6}.$$

In the first case, noting that

$$\frac{1}{4}n^{-1/2} = \min_{z \in [0,1]^n} \|z\|_2 \quad \text{subject to} \ z^T\mathbf{1} \geq \frac{1}{4},$$

we would have that $\|\Delta A\|_F \geq \|(\Delta A)_{i.}\|_2 \geq n^{-1/2}/4$. In the second case, we would have $\|\Delta b\|_2 \geq |\Delta b_i| \geq 1/6$.

Suppose that the infeasibility happens instead with respect to one of the $x \leq \mathbf{1}$ constraints. A similar argument for the violated constraint would lead to the same necessary condition (26) and the same bounds.

In either case, assuming that $n \geq 3$, we have

$$\|(\Delta A, \Delta b, \Delta c)\| \geq n^{-1/2}/4,$$

so that

$$\delta_P \geq \|d\|^{-1}n^{-1/2}/4.$$

**Dual Bound.** We consider here a fixed vector $(u, z) = 0$. For infeasibility, we would need $\Delta c_i < -1$ for some $i$, and therefore $\|\Delta d\| \geq 1$. We thus have

$$\delta_D \geq \|d\|^{-1}$$

Putting the primal and dual bounds together, and using our bound on $\|d\|$, we obtain

$$\frac{1}{\delta_P\delta_D} = O(\|d\|^2 n^{1/2}) = O((m + n)n^{1/2}).$$

### D.2 Packing and Covering Programs

Suppose we have a covering program with data $(A, b, c) \geq 0$, with $[0, 1]$ bound constraints on each variable. That is,

$$\min_{x \in \mathbb{R}^n_+} c^T x \quad \text{subject to} \ Ax \geq b, x \leq \mathbf{1},$$

its dual is a packing program:

$$\max_{u \in \mathbb{R}^m_+, z \in \mathbb{R}^n_+} u^T b - z^T\mathbf{1} \quad \text{subject to} \ A^T u - z \leq c.$$

Generalizing our argument above, we find a point that has the most slack from each constraint. Defining the following measure of slack:

$$s(A, b, c) = \max_{x \in \mathbb{R}^n_+ : Ax \geq b, x \leq \mathbf{1}} \min\{ \min_{i=1,\dots,n} 1 - x_i, \min_{j=1,\dots,m} b_j - A_j.x \},$$

we can obtain a lower bound $\delta_P \geq \|d\|^{-1} n^{-1/2} s(A, b, c)/2$, as follows. Suppose that $x_S$ is the point that achieves the maximum slack. We need that one of the following conditions holds for at least one constraint $i$: $\Delta A_i.x_S > s(P)/2$ or $|\Delta b_i| \geq s(P)/2$. Observe that

$$\Delta A_i.x_S \leq \|x_S\|_2 \|\Delta A_i.\|_2 \leq n^{1/2} \|\Delta A_i\|_2.$$

(The second inequality follows from $0 \leq x_S \leq \mathbf{1}$.) Thus, in this case, $\|\Delta A_i\|_2 > s(A, b, c) n^{-1/2}/2$. Using a similar argument to the previous subsection, we have

$$\delta_P \geq \|d\|^{-1} \|\Delta d\| \geq \|d\|^{-1} s(A, b, c) n^{-1/2}/2.$$

Since $(u, z) = (0, 0)$ is feasible for the dual, we have by a similar argument to the previous subsection that infeasibility occurs only if $|\Delta c_i| \geq c_i$ for at least one $i$. We therefore have $\|\Delta d\| \geq \min_{i=1,2,\dots,n} c_i$, so that

$$\delta_D \geq \|d\|^{-1} \min_{i=1,2,\dots,n} c_i.$$

Putting the bounds on $\delta_P$ and $\delta_D$ together, we have

$$\frac{1}{\delta_P \delta_D} \leq \|d\|^2 \frac{1}{s(A, b, c) \min_{i=1,2,\dots,n} c_i} O(n^{1/2}).$$

## E   Extended Experimental Results

In this section, we elaborate our discussion on the experimental results in Section 4.2 and provide additional evidence to support our claims. Figures 5 and 6 compare the performance of Thetis with Cplex-IP and Cplex-LP on all tested instances of vertex cover, independent set, and multiway-cut. In all three formulations, we used unit costs in the objective function. The results in Figure 6 were obtained by using default tolerance on Cplex-LP, while Figure 5 uses the same tolerance setting as the main manuscript.

**Maximum Independent Set.**   We observed that the rounded feasible solutions obtained using Thetis were of comparable quality to those obtained by rounding the more accurate solutions computed by Cplex-LP. The integral solutions obtained from Cplex-IP were only marginally better than that obtained by LP-rounding, but at a cost of at least an order of magnitude more time.

**Multiway Cuts.**   The number of variables in the multiway-cut problem is $O((|E| + |V|) \times k)$ where $|E|$ is the number of edges, $|V|$ is the number of vertices and $k$ is the number of terminals. The terminals were chosen randomly to be in the same connected component of the graph. All the test instances, excepting Google+, were fully connected. For Google+, 201949 (of 211186 vertices) were connected to the terminals. For all instances, including Google+, all codes were run on (20) built using the entire graph.

We solved the QP-approximation of (20) using a block-SCD method, which is variant of Algorithm 1, in which an update step modifies a block of co-ordinates of size $k$. For the blocks corresponding to variables $x_v$, $\forall v \in V$, we performed a projection on to the $k$-dimensional simplex $\Delta_k$. The simplex projection was necessary to ensure that the approximate LP solution is always feasible for (20). We disabled presolve for Thetis to prevent the simplex constraints from being eliminated or altered. We did not disable presolve for Cplex-LP or Cplex-IP.

Our results demonstrate that Thetis is much more scalable than both Cplex-IP and Cplex-LP. Thetis was an order of magnitude faster than Cplex-LP on the Bhoslib instances while generating solutions of comparable quality. Both Thetis and Cplex-LP recovered the optimal solution on some of the instances. On the SNAP instances, both Cplex-IP and Cplex-LP failed to complete within an hour on any of the instances. Cplex-IP was able to generate feasible solutions using its heuristics, but was able to unable to solve the root-node relaxation on any of the SNAP instances.

| VC (min) | Cplex IP | | | Cplex LP | | | Thetis | | |
|---|---|---|---|---|---|---|---|---|---|
| | t (secs) | BFS | Gap(%) | t (secs) | LP | RSol | t (secs) | LP | RSol |
| frb59-26-1 | - | 1475 | 0.7 | 2.48 | 767.0 | 1534 | 0.88 | 959.7 | 1532 |
| frb59-26-2 | - | 1475 | 0.6 | 3.93 | 767.0 | 1534 | 0.86 | 979.7 | 1532 |
| frb59-26-3 | - | 1475 | 0.5 | 4.42 | 767.0 | 1534 | 0.89 | 982.9 | 1533 |
| frb59-26-4 | - | 1475 | 0.5 | 2.65 | 767.0 | 1534 | 0.89 | 983.6 | 1531 |
| frb59-26-5 | - | 1475 | 0.5 | 2.68 | 767.0 | 1534 | 0.90 | 979.4 | 1532 |
| Amazon | 85.5 | $1.60\times10^5$ | - | 24.8 | $1.50\times10^5$ | $2.04\times10^5$ | 2.97 | $1.50\times10^5$ | $1.97\times10^5$ |
| DBLP | 22.1 | $1.65\times10^5$ | - | 22.3 | $1.42\times10^5$ | $2.08\times10^5$ | 2.70 | $1.42\times10^5$ | $2.06\times10^5$ |
| Google+ | - | $1.06\times10^5$ | 0.01 | 40.1 | $1.00\times10^5$ | $1.31\times10^5$ | 4.47 | $1.00\times10^5$ | $1.27\times10^5$ |
| **MC (min)** | Cplex IP | | | Cplex LP | | | Thetis | | |
| | t (secs) | BFS | Gap(%) | t (secs) | LP | RSol | t (secs) | LP | RSol |
| frb59-26-1 | 72.3 | 346 | - | 312.2 | 346 | 346 | 5.86 | 352.3 | 349 |
| frb59-26-2 | 561.1 | 254 | - | 302.9 | 254 | 254 | 5.82 | 262.3 | 254 |
| frb59-26-3 | 27.7 | 367 | - | 311.6 | 367 | 367 | 5.86 | 387.7 | 367 |
| frb59-26-4 | 65.4 | 265 | - | 317.1 | 265 | 265 | 5.80 | 275.7 | 265 |
| frb59-26-5 | 553.9 | 377 | - | 319.2 | 377 | 377 | 5.88 | 381.0 | 377 |
| Amazon | - | 12 | NA | - | - | - | 55.8 | 7.3 | 5 |
| DBLP | - | 15 | NA | - | - | - | 63.8 | 11.7 | 5 |
| Google+ | - | 6 | NA | - | - | - | 109.9 | 5.8 | 5 |
| **MIS (max)** | Cplex IP | | | Cplex LP | | | Thetis | | |
| | t (secs) | BFS | Gap(%) | t (secs) | LP | RSol | t (secs) | LP | RSol |
| frb59-26-1 | - | 50 | 18.0 | 4.65 | 767 | 15 | 0.88 | 447.7 | 18 |
| frb59-26-2 | - | 50 | 18.0 | 4.74 | 767 | 17 | 0.88 | 448.6 | 17 |
| frb59-26-3 | - | 52 | 13.4 | 3.48 | 767 | 19 | 0.87 | 409.2 | 19 |
| frb59-26-4 | - | 53 | 11.3 | 4.41 | 767 | 18 | 0.90 | 437.2 | 17 |
| frb59-26-5 | - | 51 | 15.6 | 4.43 | 767 | 18 | 0.88 | 437.0 | 18 |
| Amazon | 35.4 | $1.75\times10^5$ | - | 23.0 | $1.85\times10^5$ | $1.56\times10^5$ | 3.09 | $1.73\times10^5$ | $1.43\times10^5$ |
| DBLP | 17.3 | $1.52\times10^5$ | - | 23.2 | $1.75\times10^5$ | $1.41\times10^5$ | 2.72 | $1.66\times10^5$ | $1.34\times10^5$ |
| Google+ | - | $1.06\times10^5$ | 0.02 | 44.5 | $1.11\times10^5$ | $9.39\times10^4$ | 4.37 | $1.00\times10^5$ | $8.67\times10^4$ |

Figure 5: Wall-clock time and quality of fractional and integral solutions for three graph analysis problems using Thetis, Cplex-IP and Cplex-LP. Each code was given a time limit of one hour, with '-' indicating a timeout. BFS is the objective value of the best integer feasible solution found by Cplex-IP. The gap is defined as (BFS−BB)/BFS where BB is the best known solution bound found by Cplex-IP within the time limit. A gap of '-' indicates that the problem was solved to within 0.01% accuracy and NA indicates that Cplex-IP was unable to find a valid solution bound. LP is the objective value of the LP solution, and RSol is objective value of the rounded solution.

| VC | Cplex-IP | | | Cplex-LP (default tolerances) | | | Thetis | | |
|---|---|---|---|---|---|---|---|---|---|
| (min) | t (secs) | BFS | Gap(%) | t (secs) | LP | RSol | t (secs) | LP | RSol |
| frb59-26-1 | - | 1475 | 0.7 | 4.59 | 767.0 | 1534 | 0.88 | 959.7 | 1532 |
| frb59-26-2 | - | 1475 | 0.6 | 4.67 | 767.0 | 1534 | 0.86 | 979.7 | 1532 |
| frb59-26-3 | - | 1475 | 0.5 | 4.76 | 767.0 | 1534 | 0.89 | 982.9 | 1533 |
| frb59-26-4 | - | 1475 | 0.5 | 4.90 | 767.0 | 1534 | 0.89 | 983.6 | 1531 |
| frb59-26-5 | - | 1475 | 0.5 | 4.72 | 767.0 | 1534 | 0.90 | 979.4 | 1532 |
| Amazon | 85.5 | $1.60 \times 10^5$ | - | 21.6 | $1.50 \times 10^5$ | $1.99 \times 10^5$ | 2.97 | $1.50 \times 10^5$ | $1.97 \times 10^5$ |
| DBLP | 22.1 | $1.65 \times 10^5$ | - | 23.7 | $1.42 \times 10^5$ | $2.07 \times 10^5$ | 2.70 | $1.42 \times 10^5$ | $2.06 \times 10^5$ |
| Google+ | - | $1.06 \times 10^5$ | 0.01 | 60.0 | $1.00 \times 10^5$ | $1.30 \times 10^5$ | 4.47 | $1.00 \times 10^5$ | $1.27 \times 10^5$ |
| **MC** | Cplex-IP | | | Cplex-LP (default tolerances) | | | Thetis ($\epsilon = 0.1$) | | |
| (min) | t (secs) | BFS | Gap(%) | t (secs) | LP | RSol | t (secs) | LP | RSol |
| frb59-26-1 | 72.3 | 346 | - | 397.9 | 346 | 346 | 5.86 | 352.3 | 349 |
| frb59-26-2 | 561.1 | 254 | - | 348.1 | 254 | 254 | 5.82 | 262.3 | 254 |
| frb59-26-3 | 27.7 | 367 | - | 386.6 | 367 | 367 | 5.86 | 387.7 | 367 |
| frb59-26-4 | 65.4 | 265 | - | 418.9 | 265 | 265 | 5.80 | 275.7 | 265 |
| frb59-26-5 | 553.9 | 377 | - | 409.6 | 377 | 377 | 5.88 | 381.0 | 377 |
| Amazon | - | 12 | NA | - | - | - | 55.8 | 7.28 | 5 |
| DBLP | - | 15 | NA | - | - | - | 63.8 | 11.70 | 5 |
| Google+ | - | 6 | NA | - | - | - | 109.9 | 5.84 | 5 |
| **MIS** | Cplex-IP | | | Cplex-LP (default tolerances) | | | Thetis ($\epsilon = 0.1$) | | |
| (max) | t (secs) | BFS | Gap(%) | t (secs) | LP | RSol | t (secs) | LP | RSol |
| frb59-26-1 | - | 50 | 18.0 | 4.88 | 767 | 16 | 0.88 | 447.7 | 18 |
| frb59-26-2 | - | 50 | 18.0 | 4.82 | 767 | 16 | 0.88 | 448.6 | 17 |
| frb59-26-3 | - | 52 | 13.4 | 4.85 | 767 | 16 | 0.87 | 409.2 | 19 |
| frb59-26-4 | - | 53 | 11.3 | 4.67 | 767 | 15 | 0.90 | 437.2 | 17 |
| frb59-26-5 | - | 51 | 16.6 | 4.82 | 767 | 16 | 0.88 | 437.0 | 18 |
| Amazon | 35.4 | $1.75 \times 10^5$ | - | 25.7 | $1.85 \times 10^5$ | $1.58 \times 10^5$ | 3.09 | $1.73 \times 10^5$ | $1.43 \times 10^5$ |
| DBLP | 17.3 | $1.52 \times 10^5$ | - | 24.0 | $1.75 \times 10^5$ | $1.41 \times 10^5$ | 2.72 | $1.66 \times 10^5$ | $1.34 \times 10^5$ |
| Google+ | - | $1.06 \times 10^5$ | 0.02 | 68.8 | $1.11 \times 10^5$ | $9.40 \times 10^4$ | 4.37 | $1.00 \times 10^5$ | $8.67 \times 10^4$ |

Figure 6: Wall-clock time and quality of fractional and integral solutions for three graph analysis problems using Thetis, Cplex-IP and Cplex-LP (run to default tolerance). Each code was given a time limit of one hour, with '-' indicating a timeout. BFS is the objective value of the best integer feasible solution found by Cplex-IP. The gap is defined as (BFS−BB)/BFS where BB is the best known solution bound found by Cplex-IP within the time limit. A gap of '-' indicates that the problem was solved to within $0.01\%$ accuracy and NA indicates that Cplex-IP was unable to find a valid solution bound. LP is the objective value of the LP solution, and RSol is objective value of the rounded solution.

## Footnotes

[5]Note that Renegar appears to use a different formulation for the linear program, namely $Ax \le b$ rather than $Ax = b$. However, his inequality represents a complete ordering with respect to a closed convex cone $C_Y$, and when we set $C_Y = \{0\}$, we recover $Ax = b$.