[Reviews · NeurIPS 2013]

Submitted by Assigned_Reviewer_4

Summary: The paper proposes a new efficient method to obtain approximate solutions to LP relaxations of integer linear programs. It argues that since the fractional solutions of the relaxation would need to be rounded in order to obtain the final solution, we could potentially get similar accuracy by rounding an approximate solution instead of the optimal one. To obtain an approximate solution, it solves a related "regularized quadratic penalty" formulation of the LP relaxation using an efficient stochastic coordinate descent algorithm. Experiments performed on tasks such as vertex cover, maximum independent set and multiway cut show that rounding the approximate solution provides similar accuracy and a decent speed-up compared to rounding the optimal solution.

Quality: Interesting paper. The algorithm appears technically correct but I didn't check the proofs thoroughly. The experiments are designed to test the main hypothesis that it is sufficient to compute an efficient, approximate solution of the relaxation.

Clarity: The paper is mostly clear. There are a large number of minor typos (some examples below). Perhaps the paper was not proof-read before submission. These can be easily corrected in subsequent versions.
- line 47: replace "specific form type" by "specific form" or "specific type"
- line 51: replace "approaches tasks" by "approaches for tasks"
- line 69: replace "to to" by "to"
- line 96: replace "Consider a graph G(V,E) denote a graph" by "Let G(V,E) denote a graph"

Originality: To the best of my knowledge, the proposed approach is new. The hypothesis that approximate solutions are sufficient for accurate rounding also appears to be new to this paper, at least within the ML community.

Experiments: The simple baseline of increasing the tolerance \epsilon for the Cplex-LP solver is not tested. Instead, the same value of \epsilon is used for both the Cplex-LP solver and the QP approximation. Is there any empirical evidence that the proposed method would be significantly faster than this simple baseline?

It would also be helpful to see how the method compares to specialized LP solvers for specific problems, for example, the LP solvers for MAP estimation.
Summary: Interesting paper. I'm leaning towards accept. I hope the authors can provide a comparison with the simple baseline during the rebuttal phase.

Post-rebuttal comments: The rebuttal has answered my only (minor) concern regarding the choice of the tolerance values. If there is sufficient space for it, please include the experiments in the final version of the paper, or at least in a related technical report.

Submitted by Assigned_Reviewer_5

Summary:
The paper considers LP relaxations in machine learning and studies the problem
of solving the LP relaxation only approximately and derives bounds on the
overall approximation rate when solving the integer program (by rounding the LP
solution).

Pros:
- very interesting problem (solve LP relaxation, knowing that in the end we will
round the solution). This is a relevant problem in many machine learning
fields.
- to the best of my knowledge the first work that gets approximation guarantees
for the overall algorithm (LP approximation by smoothing, only solve smoothed
LP to a some threshold, integer rounding).

Cons:
- I would skip the SCD algorithm in the paper and rather try to give more
intuition and proof sketches in the main part of the paper. I do not think a
reader can appreciate section 3.3 too much. It looks too me that the running
time is not very exciting, both n and 1/epsilon^2 appear (SGD does for example
not have the dependence on n in SVMs), why not solve the problem with a more
standard solver? It seems to me SCD is not the main contribution of the paper.
- the speed improvements seen in the experiments are not hugely impressive
(roughly one order of magnitude). Also there is probably some precision option
in CPLEX, what happens if you lower the precision in CPLEX LP?
- few language problems, nothing major.
- long appendix

Quality:
Experiments could be improved, some questions about the main theorems remain.

Clarity:
Language problems and suboptimal choices in the presentation (e.g. get rid of
3.3 and include more material from the appendix).

Originality:
To the best of my knowledge the main section (3.2) is original.

Significance:
LP relaxations are important tools for combinatorial optimization in machine
learning.

Comments:
39: Despite ?
40: .. Gurobi cannot scale .., the order 10 of speedup of your method is
probably not going to change this.
43: -- but often much more efficiently. Rewrite.
52: Cplex do no scale to
52: The Cplex LP solver was slower but it ran through in the time limit on
almost all problems.
55: Obviously message passing is related in that it also solves LP relaxations, but I fail to
see the connection to your method. The smoothing?
69: in which to articulate
74: would appreciate some citations
98: is a subset of V each edge (rephrase!)
102: vertex cover as a
104: for each vertex v \forall v \in V, rephrase!
130: one can easily
132: and an example such scheme (rephrase)
144: Figure caption needs more details about the parameters.
(3): min_x
(4): max_u
167: is the definition standard? if so, cite
177: \prod_[0,1]^n was not really introduced
176: to the linear program in
193: how is d defined, it is a concatenation of vectors and a matrix, what's the
norm on d?
191: the paragraph here seems to apply to general LPs, it is a bit confusing, as
it's introduced after (5)
236: \beta depends on optimal solution, how would we estimate this?
280: algorithm 2 is described (rephrase)
304: what does thetis stand for?
310: have rarely been used? for
315: Boolean
317: the resulting graphs
320: Gibbs sampling is not really state of the art. How exactly, annealing?
321: why didn't you also use an LP MAP solver such as the one in [14]?
331: have placed if (rephrase)
Figure 2: what about energies (apart of P, R, F1)?
357: in this paragraph you mention the appendix twice
360: the second family of
369: this experiment is to compare
418: Cplex-IP on large scale
960: describe difference between Table 3 + 4 in more detail
Summary: Significant work, but I feel the writeup is lacking in many parts and the
experiments could be improved.

Submitted by Assigned_Reviewer_6

Paper summary.
The authors propose an efficient scheme to solve LP relaxations of
combinatorial optimization problems. Their contribution is a novel scheme and
analysis that takes into account the original goal of constructing an integral
feasible solution from the relaxed solution. They prove that an approximate
solution is sufficient to construct an integral alpha-approximate solution to
the vertex cover problem. They also prove a convergence result for an
algorithm solving a suitably constructed QP approximation to a general
standard form LP problem. The proposed method is evaluated experimentally on
a number of combinatorial optimization problems and shown to be competitive
with Cplex, a state-of-the-art LP solver.


Review.
The paper is well written and the technical contributions are clear and well
motivated. I like the general direction of tailoring the solver of the
relaxation for the final goal of constructing good primal feasible solutions
and I believe the authors make valuable contributions in this direction.

The technical results make sense but I did not check carefully the
derivations. The dependence on the well-conditioning of the linear program
(\delta_P and \delta_D) seems useful for deriving the results but it is not
clear how these numbers and the proposed method would behave for more general
problems, say MAP-MRF LOCAL relaxations; in particular we know that modern LP
solvers such as Cplex and Gurobi are exceptionally robust to ill-conditioning.

The experiments are a mixed bag. It seems (Table 1) that the Cplex-LP results
are not comparable to the results by the proposed method: the speed up (as
measured by S) is provided by the proposed method but the solutions are very
much inferior (as measured by Q). As such it is not clear to me that one is
better than the other. The particular results in Table 1 (and Table 2) are
not referenced explicitly from the main text, but at least from Table 1 I
would disagree about the proposed method being comparable to Cplex (which is a
big ask, I know).

The claim that Cplex and Gurobi are not competitive for solving linear
programs arising from machine learning applications could be toned town a
little; see the recent study (Kappes et al., "A Comparative Study of Modern
Inference Techniques for Discrete Energy Minimization Problems", CVPR 2013),
which has shown that Cplex is often providing exceptional performance for
problems with up to a hundred thousand variables (a size comparable to many of
the problems solved in the submitted paper).

Overall I liked the paper and believe the contribution to be an important one.


Minor details.
Table 1 caption: please say "lower values of Q are better" and "higher values
of Q are better", as the previous sentence mentions both S and Q.
Reference [15], "Mathematical".
Summary: A well written paper with an interesting direction and solid technical
contributions.
Author Feedback

Author rebuttal: We thank the reviewers for their thorough reports and stimulating comments.

Reviewers 4 and 5 raised concerns about the many typographical and presentation errors. We are grateful that they persevered with their reviews despite these errors, which are being fixed as we revise the manuscript.

TO REVIEWER 4: Our experiments use comparable (fairly loose) tolerances for both Cplex-LP and Thetis solvers. Cplex is relatively insensitive to tolerance, while Thetis is faster at finding less accurate solutions. For both methods, the quality of integral solutions are not affected by the choice of tolerance. These observations are confirmed by the following table, which reports on the frb59-26-1 instance for the vertex cover problem for three values of epsilon: 10^{-1,-3,-5}.

Table: Quality and computational time of fractional and integral solutions for 3 values of epsilon for vertex cover on frb59-26-1.
Thetis Cplex-LP
| 1e-1 1e-3 1e-5 | 1e-1 1e-3 1e-5 |
Fractional Solution | 768.2 767.2 767 | 767 767 767 |
Integral Solution | 1534 1533 1533 | 1534 1534 1534 |
Solve time (s) | 0.65 6.89 14.33 | 5.06 4.48 4.47 |

Specialized LP solvers for MAP estimation: We have not yet had time to implement the algorithm from [14], but we repeated the experiment in Figure 2 using Cplex-IP. The following table shows quality metrics of the optimal IP solution of MAP estimation. We refer to Figure 2 of the paper for details of notation.
Cplex-IP
Dataset | P R F1 |
CoNLL | .87 .91 .89 |
TAC-KBP | .80 .80 .80 |

The solutions obtained using Thetis are of similar quality.

TO REVIEWER 5: We believe that the SCD method for QP is vital to our approach, as it allows low-accuracy solutions of the QP to be obtained
quickly. By contrast, the runtime of interior-point QP methods is not very sensitive to convergence tolerance. Moreover, an efficient
multicore implementation of SCD is easy, whereas parallelization of simplex and interior-point methods is much harder.

The analysis in Sections 3.2 and 3.3 allows us to close the loop: we are able to derive a worst-case complexity bound for the Thetis
approach.

Comparisons with stochastic gradient (SG): To obtain the desired accuracy in constraints and function value of the LP - captured by epsilon - we need to solve the QP to within a different, tighter tolerance, quantified by bar{epsilon}. Both tolerances are related to the choice of penalty parameter beta in the QP. Ignoring here the dependence on dimensions, we note the relationships beta ~ epsilon^{-1} (Theorem 5) and bar{epsilon} ~ beta^{-3} ~ epsilon^3. Expressing all quantities in terms of epsilon, and using Theorem 6 and subsequent discussion, we see an iteration complexity of epsilon^{-2} for SCD (ignoring log terms). (The linear convergence rate of SCD is instrumental to this favorable value.) By contrast, standard variants of SGD applied to the QP yield poorer complexity. For diminishing-step or constant-step variants of SG, we see complexity of epsilon^{-7}, while for robust SG, we see epsilon^{-10}. (Besides the inverse dependence on bar{epsilon} or its square, there is a contribution of order epsilon^{-2} from the conditioning of the QP. Further details will appear in the revision; )

As to dimension dependence: we note that beta depends on dimension (see e.g. lines 234-236) so that n and m enter into the SG complexity, as they do into SCD. We will investigate dimension dependence further.

We agree with the reviewer that high-quality QP solvers are available, but we believe that SCD has an advantage in obtaining low-accuracy solutions and in multicore implementation. A simple experiment compares Cplex's QP solver with SCD on equation 5 of our paper, arising from a vertex cover LP, setting beta=0.1 and solving to an accuracy of 10^{-3}. The resulting QP may be harder than the original LP. On these tests, SCD is two orders of magnitude faster.
Time (s)
Dataset | Cplex-QP | SCD |
Amazon | 1435.38 | 8.42 |
DBLP | 560.30 | 5.85 |
Google+ | - | 10.43 |
('-' indicates timeout after 1 hr).

Thetis vs Cplex-LP run time: On some instances we outperform Cplex by more than an order of magnitude. More importantly, we can obtain
results in cases where Cplex is not able to execute at all.

Prompted by the referee, we clarify our experimental methodology. Our instances to fall in to three categories:
(1) All methods could run;
(2) Cplex-LP could run, but not Cplex-IP;
(3) Only Thetis works.
In addition, we selected standard benchmark instances. We will study further the way in which 'real-world' problems are distributed among these categories.

The reviewer raises the possibility of speeding the computation by loosening the Cplex LP tolerance in solving the LP approximation. In resonse, we point to Tables 3 and 4 of the appendix, which repeats the experiment in Table 2 of the main manuscript by setting eps=0.1 and the default tolerance of Cplex-LP

TO REVIEWER 6: We agree that the paper should tone down its claim that Cplex and Gurobi are unsuited to machine learning applications, and we are grateful for the reference provided. We will make the more limited claim that we identify a class of problems where using approximate LP solutions for LP-rounding is a competitive approach.

In Table 1, Q measures the quality of the solution obtained using our approach as a ratio of the solution obtained by Cplex-IP (a proxy for the true optimal solution). Speedup S measures the ratio of the time taken by Cplex-LP and our approach to obtain a fractional solution. Our approximate LP solver generates integral solutions of comparable quality with Cplex-LP. This is our final quality goal, even if the intermediate solutions are of lower quality.